# Transforming Smallholder Farmers Support with an AI-Powered FAQbot: A Comparison of Techniques

## Abstract

Access to sufficient information on desired agricultural practices, such as planting period, when to apply fertiliser, how to transport grains, etc. is of utmost importance in the agricultural industry as it directly affects farm yields. The responses to these questions are closed domain, therefore leading to the development of a question-answering conversational bot (FAQbot) that can provide the appropriate responses immediately. This study undertakes a comparative analysis of three distinct methodologies for constructing a FAQbot. These approaches encompass the development of a generative-based chatbot employing BERT and GPT-2, the creation of an intent classification model leveraging PyTorch and the Natural Language Toolkit (NLTK) libraries, and the implementation of an information retrieval-based model utilising pre-trained Large Language Models (LLMs) using Langchain. Our methodological framework includes the transformation of a FAQ dataset into formats suitable for chatbot training, specifically CSV and JSON. Notably, the retrieval-based method surpassed the generative-based and intent classification methods by consistently providing precise answers for every question in the database, irrespective of rephrasing or reframing.

Keywords: Agriculture, FAQBot, LLMs, Natural Language Processing

## 1 Introduction

The agricultural sector is a treasure trove of valuable information, but accessing it can be a daunting challenge. This wealth of knowledge is often scattered across research papers, news articles, and specialised journals, and what makes it even more complex is the sector's highly domain-specific nature. Agriculture is not a one-size-fits-all field; it is deeply influenced by factors like climate change, geographical location, seed quality, and the use of fertilisers. These variables underscore the critical importance of farmer's access to precise information that aligns with their unique environmental conditions, as this directly and profoundly impacts their crop yields.

In the realm of domain-specific information, there is a pressing need for both retrieval and generation of closed-domain knowledge. R. Dsouza & Kalbande (2019) distinguish between two primary design approaches for chatbots: retrieval-based and generative-based systems. Retrieval-based chatbots excel at providing responses that closely align with the content present in their pre-existing database of responses, resulting in coherent appropriate answers. Information Extraction (IE) plays a pivotal role in this context, representing a methodical process aimed at discerning and pulling out pertinent data from a corpus of documents or textual sources Yanshan et al. (2018). This operation is triggered by a user's query and encompasses the identification of specific entities, relationships, attributes, or facts that directly address the user's question or inquiry.

In contrast, generative-based chatbots adopt a different strategy, harnessing their capacity to learn and adapt by drawing from a diverse array of conversational interactions, enabling them to craft responses that transcend the limitations of a fixed response bank. For generative models, the responses are crafted based on the knowledge accumulated from the training data and the provided context. To navigate the challenges posed by limited data availability, an alternative strategy was explored: intent classification. This technique harnesses feed-forward neural networks to predict the underlying

intent behind user questions, allowing for more targeted responses even without extensive training data.

Before LLMs like GPT-3, traditional methods in natural language processing relied on rule-based systems, manual annotation, and domain-specific tools like Named Entity Recognition (NER) and keyword-based information retrieval. Chatbots started with being rule-based; they were programmed to give specific responses to inputs based on predefined rules; however, they were limited in that they could not handle ambiguity and were not flexible enough. In the early 1990s, statistical methods were introduced. It focused on predicting the probability of a word given its context. However, it was limited due to data sparsity, as there are numerous combinations of words it is not exposed to, therefore not being able to understand its context Manish (2023). The introduction of deep learning was a game changer, from feed-forward networks to recursive networks. It had a significant improvement in pattern recognition, and high performance as it could learn from large-scale data StackRoute (2023). LLMs like LLaMA and ChatGPT introduced a shift by leveraging pre-trained models to handle various tasks and domains, reducing manual effort.

Recent research, including studies by Monica et al. (2021), highlights LLMs' superiority over traditional methods in Information Extraction (IE). However, LLMs face challenges like hallucinations and memory efficiency issues. To address these limitations, Retrieval-based question answering models retrieve relevant documents or passages from a database and extract information to provide accurate answers Simran et al. (2023).

## 2 METHODOLOGY

In this section, we provide a detailed account of the implementation procedures associated with the three approaches employed in constructing our FAQbot.

### 2.1 DATA PREPROCESSING

The study's dataset comprises question-answer pairs extracted from FAQs of an agricultural firm, sourced from a CSV database containing around 38 pairs. Each pair is associated with a specific tag, enabling coherent categorisation of questions into functional categories, yielding approximately 32 unique tags. The CSV file suitable for the retrieval and generative-based approaches has three columns: "questions," "answers," and "tags", while the JSON format works for the intent classification approach. General preprocessing steps were performed including special character removal using regular expressions and word decontraction.

For the generative-based approach, we require both positive and negative samples of question-answer pairs. Positive samples consist of question Q and answer A from the same pair, labelled 1.0. Negative samples, labelled as -1.0, comprise an original question Q and an answer A' from the dataset, where the tag T' differs from the tag T associated with the original question. In total, there are 38 positive pairs and 36 negative pairs.

### 2.2 GENERATIVE-BASED MODEL APPROACH

For this approach, we employed supervised training with a sequence-to-sequence (seq2seq) framework, harnessing the capabilities of pre-trained models like BERT and GPT-2. This approach, inspired by the GitHub repository `Doc Product: Medical Q&A with deep learning models` Santosh et al. (2022), unfolds in two stages.

#### 2.2.1 BERT FINE TUNING AND TRAINING EMBEDDING EXTRACTOR

In this stage, we utilise a pre-trained BERT base-cased model and fully connected neural networks (FCNN) to extract embeddings from question-answer pairs in the training dataset. These embeddings are used for semantic search using Facebook AI's similarity search tool (FAISS). The BERT model encodes the question-answer pairs separately, passing their outputs through two distinct FCNNs to obtain embeddings.

During training, we employ a Mean Squared Error (MSE) loss function to measure the difference between expected cosine similarities (ranging from -1.0 to 1.0) and those predicted by our model.

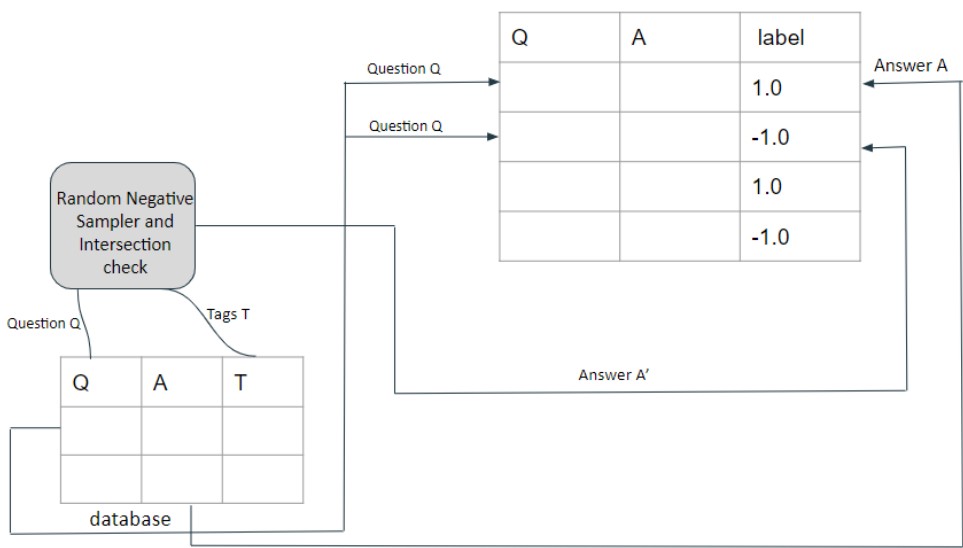

Figure 1: Creation of negative points data

For model evaluation on the validation dataset, we analyse cosine similarities for correctly classified negative and positive instances, comparing them to those of incorrectly classified instances.

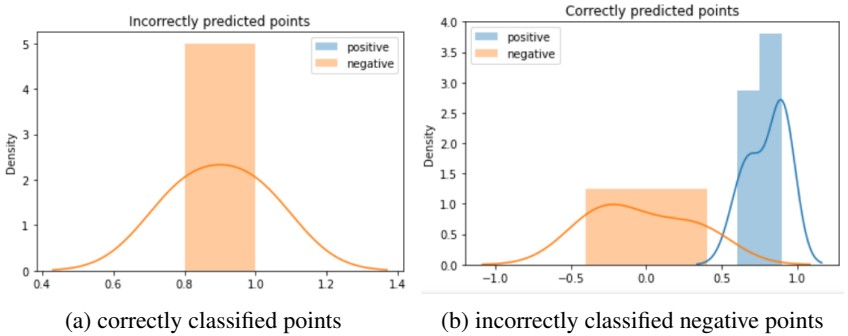

(a) correctly classified points          (b) incorrectly classified negative points

Figure 2: Plots of predicted cosine similarities

From Figure 2, it is evident that all positive points are correctly classified, but there are misclassified negative points. Further analysis reveals that a 0.5 threshold effectively separates positive from negative points. Using this threshold, our model achieved an 87% accuracy rate. Following training, we save the extracted question and answer embeddings in a CSV file in NumPy array format.

### 2.2.2 FINE-TUNING GPT-2 MODEL

After extracting question and answer embeddings, our system conducts semantic searches for each question, retrieving similar answers and their corresponding questions. This search employs cosine similarity calculations via FAISS, comparing the current question's embeddings to those of stored answers. FAISS generates sorted pairs of similar question-answer pairs (e.g., Q1A1, Q2A2, Q3A3) related to the queried question.

Next, we fine-tune the GPT-2 model to generate answers for the questions. The model takes concatenated sequences, arranged as (Q3A3Q2A2Q1A1Q). This arrangement ensures that the most relevant question-answer pair is closest to the question (Q). Our pre-trained GPT-2 model can handle a maximum sequence length of 1024 tokens, so after tokenization, we keep the last 1024 tokens. This choice ensures that crucial context is retained, with any omitted information coming from less sim-

ilar question-answer pairs. The GPT-2 model's output follows the format Q3A3Q2A2Q1A1QA', where A' represents the generated answer for the given question.

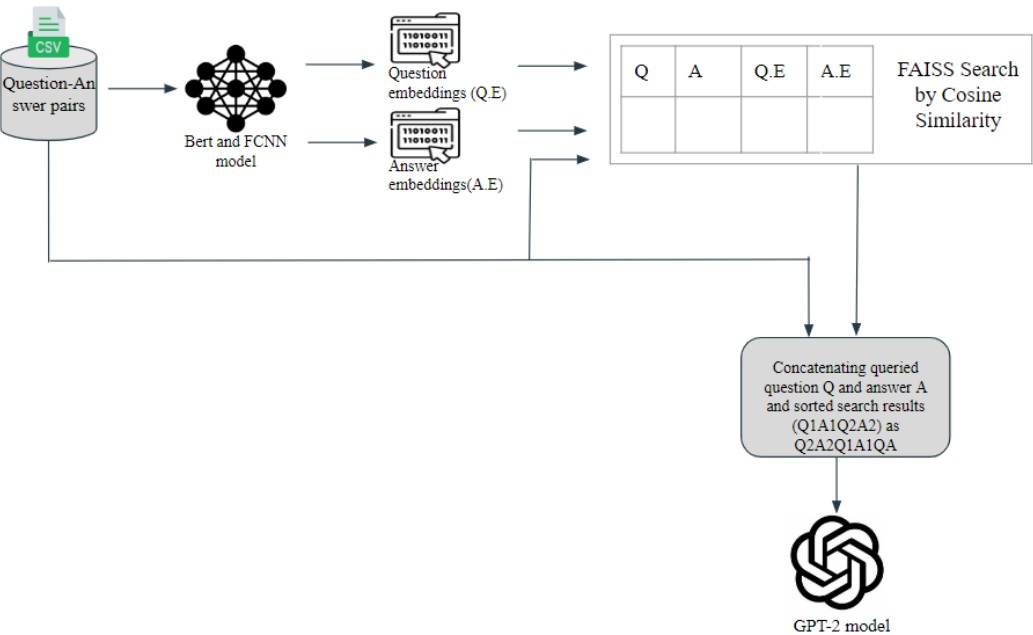

Figure 3: Flow-chart for generative-based model approach

## 2.3 INTENT CLASSIFICATION APPROACH

For this approach, we returned to the basics of using NLTK and deep learning. NLTK has several methods and pre-built models that can be used for different text analyses. It is used here for some preprocessing steps listed below. The text processing steps utilised were;

- Tokenization: The texts were broken down into smaller functional semantic units called tokens for processing.
- Lowercase conversion: To avoid redundancy in the token list and confusing the model when it sees the same word but with different cases, we convert all the texts into lowercase.
- Stemming: Texts were normalised by reducing inflected words to their word stem.
- Bag of words (BoW): This extracted features within our texts. A list containing all unique words present in our dataset is compiled. Subsequently, we created a list of 0s by traversing this list. If a word from the list was found in the tokenized question, we replaced a corresponding '0' with '1', making our BoW.

For this approach, a feed-forward neural network was constructed utilising the PyTorch framework. The input layer's dimensions correspond to the count of distinct words in the entirety of the dataset. The hidden layers were designed with sizes that were integer multiples of 2, and the output layer was dimensioned to align with the number of distinct tags in the dataset. Rectified Linear Units (ReLU) was the activation function utilised, while a dropout layer was incorporated to mitigate overfitting concerns. The feature (input) to the model was the tokenized question as a NumPy array of the BoW, while the target (output) is the tags that were label encoded.

In the training phase, a "NeuralNet" class was designed, encompassing the architectural layers as previously delineated. Regarding model optimization, the CrossEntropyLoss function was used in conjunction with the Adam optimizer. An iterative training process ensued, during which various hyperparameters such as the batch size, and learning rate underwent tuning. For inference, a pre-determined probability threshold of 0.65, was applied to gauge the appropriateness of the predicted tag for use in retrieving responses from the database.

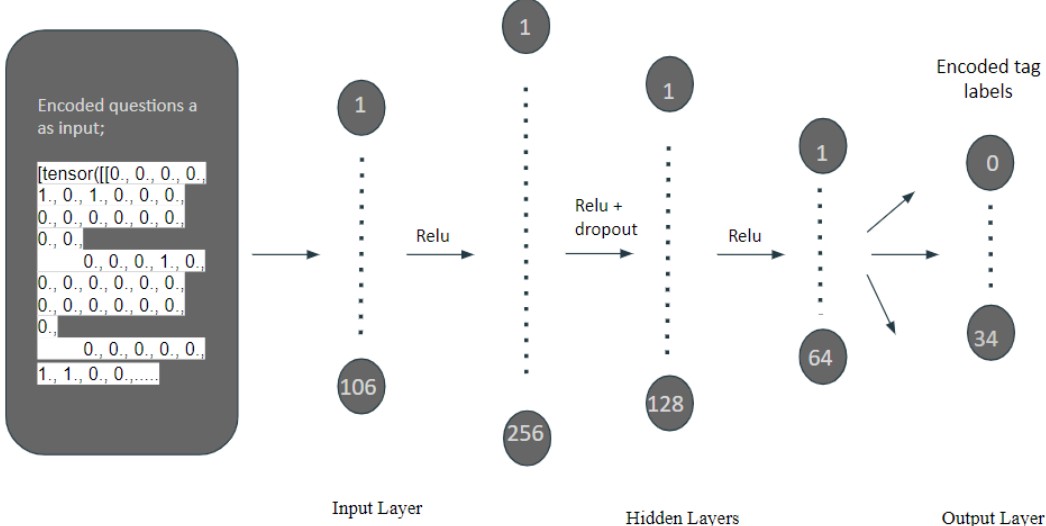

Figure 4: the feed-forward neural network architecture

## 2.4 RETRIEVAL-BASED MODEL APPROACH

The retrieval-based model approach used in this study seeks to extract exact unfiltered answer pairs to a closed-domain question, eliminating the need for phrase retrieval and document processing during inference. The diagram below shows the steps to extract answers from a question-answer (QA) pair.

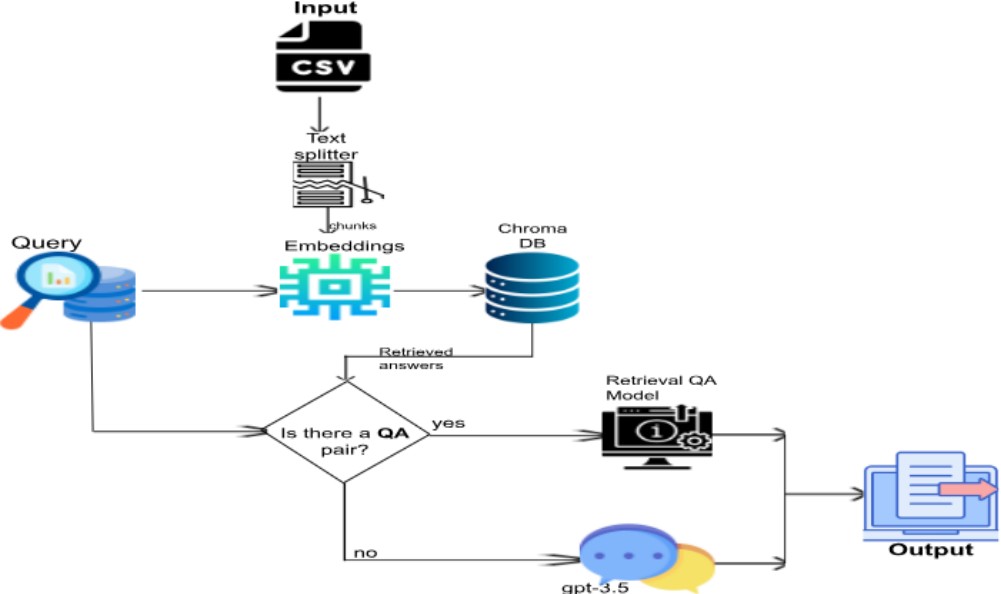

Figure 5: Diagrammatic representation of process pipeline for Retrieval Based Approach

This methodology unfolds in two phases. For the purpose of retrieval in this study, we examine the likeness between user queries (q) and the questions in our database (Q), as well as question-answer (QA) pairs. While research on community QA often employs an extensive pool of QA pairs to understand q-A relevance, we refrain from using this approach due to the relatively limited scale of our QA entries for effective model training. Our proposed method presents a fusion of the q-Q cosine

similarity and the q-A relevance gleaned from the QA pairs within our database. Subsequently, when there emerges scant or negligible similarity, we pivot to the GPT-3 model to generate answers. The methodology summarised above is further expatiated below;

### 2.4.1 EMBEDDINGS AND VECTORISATION

Before commencing the process, we accessed crucial modules within LangChain, including CSVLoader, CharacterTextSplitter, and OpenAIEmbeddings. Their roles in this process are elaborated upon below.

The procedure initiated with the loading of structured textual data via the CSVLoader module. Subsequently, the documents underwent segmentation into smaller units using the CharacterTextSplitter. To capture the semantic essence of the text, OpenAIEmbeddings were employed. These embeddings convert text into numerical representations, facilitating text similarity comparisons. Furthermore, we leveraged the Term Frequency-Inverse Document Frequency (TF-IDF) vectorizer and cosine similarity check from scikit-learn, which bolster the relevance of user queries for retrieving answers from the vector database. Notably, it's worth mentioning that CSVLoader, CharacterTextSplitter, and OpenAIEmbeddings are integral methods within the LangChain framework.

The equation below illustrates the method by which query similarities are assessed.

$$q, Q = \frac{q.Q}{||q|| \; ||Q||}$$

| | |
|---|---|
| $q$ | user's query |
| $Q$ | Question in database |
| $q.Q$ | dot product between the user's query and question in the database |
| $||Q||$ | magnitude of question vector in database |
| $||q||$ | magnitude of user's query vector |

### 2.5 DOCUMENT STORAGE AND RETRIEVAL

The preprocessed text segments undergo a transformation and are stored within a Chroma vector store. This facilitates the efficient retrieval of documents based on their proximity to the input question. A retriever interface streamlines this retrieval process, ensuring both accuracy and rapid document selection. The vector index retrieval mechanism forms the cornerstone of our information pipeline. It capitalises on the Chroma vector store, which houses the embedding vector representations of documents. This encoding method captures the semantic essence of the documents using high-dimensional vectors. In the retrieval process, the vector index retrieval employs a similarity-driven approach to pinpoint the most pertinent answers within the database. Given a query q, it retrieves a predetermined count of top k results, often with a cosine similarity score threshold of 0.7, which signifies the similarity between the query (**q**) and questions in the database (**Q**). The answer (**A**) within the **QA** pair corresponding to the most similar question is returned. The parameter k is designated within the as retrieval method, under the `search kwargs="k":<number of highest similar answer>` argument. This value can be fine-tuned to strike a balance between precision and efficiency. When presented with a user's question (**q**), the Retrieval QA model furnishes a dictionary encompassing the query, the obtained results, the source document, and metadata that pinpoints the row from which the answers were extracted. Notably, the 'result' key in this dictionary contains a substantial portion of the information found in the source documents, albeit not in its entirety. Acknowledging the vital significance of all information, it becomes imperative to return to the source documents for a comprehensive extraction of all answers.

### 2.6 CHAIN CONSTRUCTION

Once the document is embedded and indexing is completed, the retrievers and LLMs can be loaded to establish an inference Retrieval-based Question Answering (Retrieval QA) system. A Re-

trievalQA chain is established using the OpenAI language model. The RetrievalQA chain, built on pre-trained language models like BERT or RoBERTa, combines document retrieval with question-answering capabilities, delivering efficient and rapid responses. It does not generate language from scratch but ranks existing text snippets for contextually relevant answers. To create this chain, we use the from_chain_type() method to create an instance of the Retrieval QA system, specifying the OpenAI Language Model, chaining type ('stuff' for comprehensive answers retrieval), a retriever from the Chroma vector storage database, and a flag to return source documents for faithful database answers.

## 3  RESULTS AND DISCUSSION

The results from the three approaches implemented are compared and discussed in this section.

During this study, three different approaches - generative-based chatbots, intent classification using feed-forward neural network chatbots and retrieval-based chatbots were implemented. Table 1 shows the performance of the implemented model on the rephrased queries. Due to the nature of the project, its performance was determined using the manual interpretation of the quality of responses returned by each model. Our best-performing, retrieval-based approach, gave the most similar or the same responses with the answers within our database. From the results, the retrieval-based approach predicted all three responses accurately, and the intent classification predicted two out of three correctly leaving the generative-based approach, which predicted one correctly.

In Table 1, examining the model responses reveals that the retrieval-based approach outperformed others, with the intent classification approach following closely. Surprisingly, the generative-based approach performed the least. R. Dsouza & Kalbande (2019) highlights that generative models require substantial data for effective training. Considering our dataset's modest size of 38 question-answer pairs, it becomes evident that our model faced limitations due to insufficient training data. This constraint hindered consistent responses across different queries. The intent classification approach's performance was hampered by its inability to accurately predict tags for approximately 15% of the queries in the database, mainly when dealing with paraphrased queries.

The retrieval-based approach consistently delivered highly accurate responses to the queries. The fundamental reason behind this accuracy lies in the retrieval QA chain's unique methodology: it retrieves source documents rather than relying on model-generated responses, which may be occasionally summarised to enhance readability. Furthermore, this approach showcases versatility by adeptly handling scenarios where queries are absent from the database. In such cases, our conversational model, GPT-3.5, seamlessly takes the reins to generate relevant responses, ensuring a robust and adaptive user experience. By providing accurate and contextually relevant answers, our chatbot offers comprehensive and valuable assistance, making it a powerful tool for a wide range of user inquiries.

## 4  CONCLUSION

The retrieval-based approach consistently outperformed the other methods, excelling at providing responses that closely matched those in the database, regardless of variations in question phrasing. The generative-based approach's limited success can be attributed to the scarcity of robust training data. On the other hand, the intent classification approach performed commendably, closely trailing the retrieval-based method. However, retrieval-based techniques stood out for their ability to consistently deliver accurate responses, demonstrating their strength in handling diverse question formulations.

This research underscores the profound impact of LLMs and innovative retrieval methodologies within the domain of Information Extraction. We used an approach that combines cosine similarity assessments with relevance-driven retrieval techniques, thereby returning precise and contextually enriched responses to user queries. This approach effectively surmounts the limitations of generative models, offering a flexible and adaptable solution for addressing a broad spectrum of user inquiries.

Table 1: Question and Answers from the subset of the test data

| Question-Answer Pairs | | Predicted Responses | | |
|---|---|---|---|---|
| Query | Expected Answer | Generative-based | Intent-Classification | Retrieval-based |
| Please, I just lost my prepaid card, what should I do? | If you lose your prepaid card, the TGL of the trust group should visit the nearest office and our customer service team will replace the... | If you lose your prepaid card the TGL of the trust group should | If you lose your prepaid card, the TGL of the trust group should visit the nearest office and our customer service team will replace the card... | If you lose your prepaid card, the TGL of the trust group should visit the nearest office and our customer service team will replace the card... |
| I am having issues with a farm in my TG, what can I do? | Immediately go and assess the field and see what the problem is and help the member to resolve it. You should report all such issues to your field officer... | If a member's farm fails and he does not get a good yield or be able to pay back his minimum commitment The Member needs to communicate... | Immediately go and assess the field and see what the problem is and help the member to resolve it. You should report all such issues to your field officer ... | Immediately go and assess the field and see what the problem is and help the member to resolve it. You should report all such issues to your field officer ...... |
| When is it okay for a pesticide to be requested? | When you notice your maize plant has brown patches all over the green leaves, the cobs begin to rust, the plant is wilting and drying up...... | When you notice your maize plant has brown patches all over the green leaves the cobs begin to rust, the plant is wilting and drying up ...... | We recommend that our trusted members apply herbicide twice to their farms. The first herbicide application is immediately... | When you notice your maize plant has brown patches all over the green leaves, the cobs begin to rust, the plant is wilting and drying up, and holes that look 1 ...... |

## 5 FUTURE RESEARCH

Given that the quality and availability of the dataset significantly influence model performance, it is conceivable that enhancing the two less-performing models could be achieved through the incorporation of more high-quality data.

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
