# OpenReview forum: "Transforming Smallholder Farmers Support with an AI-Powered FAQbot: A Comparison of Techniques"
_ICLR.cc/2024/Conference — Submitted to ICLR 2024_

### Official Review · Reviewer_MXc3 · 2023-10-25

**Soundness:** 2 fair
**Presentation:** 2 fair
**Contribution:** 1 poor
**Rating:** 3
**Confidence:** 4

**Summary:**

The authors compare three methodologies for constructing an FAQbot for questions related to farming. They compare a generative method implemented with GPT2 and BERT, an intent classification method implemented with NLTK, and an information retrieval method using LangChain (also LLMs)

**Strengths:**

- The paper is easy to follow and well-written. The illustrative diagrams are helpful (Figs 1 and 3)
- I appreciate that the use case of a FAQbot for farms is solid, and grounded in actual user needs. Often, the NLP community spends too long evaluating on benchmark datasets that are not really related to real-world use cases, so this is a nice change.

**Weaknesses:**

- I am not sure what the overall contribution of the paper is. I might be misunderstanding, but none of these approaches seem to be new, and I am not convinced that the comparison between them would extrapolate to other scenarios
- Where are the results? Table 1 only shows 4 datapoints, but I'm assuming you evaluated on more than that. Do you have any statistics over aggregated results?
- Are the datapoints you used to evaluate just rephrasing of the training datapoints? If so, I'm not convinced that this means the system would do well on questions that are actually new.
Nit: Fig 5 looks stretched horizontally

**Questions:**

(See "Weaknesses")

---

### Official Review · Reviewer_WuHD · 2023-10-29

**Soundness:** 2 fair
**Presentation:** 2 fair
**Contribution:** 1 poor
**Rating:** 3
**Confidence:** 3

**Summary:**

The paper discusses the methods for building a FAQ chatbot in agricultural. It compares the retrieval-based and generative-based methods and provides detailed descriptions how these methods work. The qualitative analysis shows that the retrieval-based method was the best.

**Strengths:**

This paper targets a meaningful application, which will have a large impact it can be robustly usable. And, the proposed methods have relative detailed descriptions.

**Weaknesses:**

The content discussed in this paper is well-known and out-dated somehow. And, the presentation style like Fig5, 6 can be refined to a more professional manner. Generally, I'm not sure ICLR is a good venue for holding this paper.

**Questions:**

N/A

---

### Official Review · Reviewer_GDnG · 2023-11-01

**Soundness:** 2 fair
**Presentation:** 1 poor
**Contribution:** 1 poor
**Rating:** 3
**Confidence:** 4

**Summary:**

This paper describes a pipeline for a chatbot that answers queries related to farm management (a FAQbot). It compares three different approaches for implementing the system: a generative models (BERT and GPT-2), an intent classification model, and a retrieval-based model using the Langchain framework. After presenting the architectural details of each approach, the authors evaluate them on three questions based on qualitative judgements, and conclude that the retrieval-based system achieves the best performance.

**Strengths:**

The paper presents a interesting application of chatbots, to improve agricultural practices. It also describe and evaluate three different methods to implement the system.

**Weaknesses:**

The paper directly applies methods that are well-known in chatbot literature (e.g., retrieval-augmented generation with Langchain). There is limited novelty in the methods and the evaluation results are based on a very small number of samples (three questions). Furthermore, there are quality issues in the presentation, especially in the figures. For those reasons, I don't think this work delivers significant research contributions.

To improve this work, would recommend the authors:
- Increase the number test samples, so that the results have higher confidence
- Include reference for the used frameworks, e.g., NLTK
- Improve the quality of the figures (resolution and scaling are not good)

**Questions:**

What are training parameters of the intent classifier in section 2.3 (batch size, learning rate, etc)? What is the number of parameters of the GPT-2 model? This information should be included to improve reproducibility.

---

### Official Review · Reviewer_VUMf · 2023-11-02

**Soundness:** 1 poor
**Presentation:** 1 poor
**Contribution:** 1 poor
**Rating:** 1
**Confidence:** 4

**Summary:**

The paper is related to the creation of a question-answering bot in the agricultural domain. The authors of the paper have implemented and compared three different methods for creating such bots.
The first method is based on generative models. The authors divided this method into two stages:
Fine-tuning BERT with FCNN model for extraction of the question and answer embedding representations using MSE Loss. This stage stands for training data augmentation.

Fine-tuning GPT-2 to generate answers to the questions on the augmented data.
As the second method, the authors implemented an intent classifier. First, the input data was preprocessed using the NLTK library. The preprocessing included steps such as tokenization, lowercase conversion, stemming and BoW (bag-of-words) vector extraction. Furthermore, a feed-forward neural network was created that takes BoW vectors as input and predicts intent labels. The number of possible labels is 35.

The final approach is a retrieval-based architecture that splits all input data into passages, extracts embeddings from these passages and stores them in the Chroma vector store. Later, these vectors are used to extract relevant responses. The next steps are to incorporate the LangChain framework to generate the output answer based on extracted data.

As a result, the authors emphasize that the retrieval-based approach outperforms the others.

**Strengths:**

The main advantage of this work is that the authors have implemented and compared different approaches using relatively new methods and frameworks, such as LangChain, and applied them to the agricultural domain.

**Weaknesses:**

1) A major weakness is the lack of data. All experiments were based on a small dataset of only 38 samples. The final comparative study was carried out on 3 samples, which is not enough to make a decision and compare different approaches. I suppose that the in-house data could be supplemented by some external sources. By the way, there is no analysis of the existing agricultural domain datasets which could be useful to create QA-pairs synthetically.

2) Another weakness relates to the analysis. The authors haven't provided any metrics or a comparative study design. For example, how many people chose the best method? What about their age, occupation, etc.? There is also no comparison with other state-of-the-art QA methods. Besides, no baseline was described.

3) The authors mentioned that the expansion of data may lead to a better quality of the generative approaches. At the same time, only the GPT-2 model was observed as a language model with full fine-tuning, and I think it's better to apply augmented data to more complex and modern models such as LLaMA2, Falcon, Mistral, etc. Finally, there are a number of different parameter-efficient fine-tuning methods, such as LoRA, P-tuning, which could be useful even for small data without full fine-tuning of model weights.
Figures 4 and 5 are not very well done. The numbers in Figure 4 are not centered. The quality of the icons in Figure 5 is low and they are compressed.

4) In addition, there are no references for some packages and models such as NLTK, LangChain, LLaMA, FAISS, etc.

**Questions:**

The third method (retrieval-based) inherently is quite close to the Retrieval-Augmented Generation approach, and I strongly recommend authors familiarize themselves with this method.

---

### Meta-Review · Area_Chair_HcGH · 2023-12-05

**Metareview:**

The paper compares three conversational QA chatbots in agricultural domain (which is an interesting domain). However, the paper has many issues (The paper is on a very small scale data, the paper is using outdated models (BERT, NLTK), figures are not clear, lack of quantitative results) and do not have strong research contributions.

**Justification For Why Not Higher Score:**

The paper has many issues, please see my meta review as well as the reviews.

**Justification For Why Not Lower Score:**

N/A

---

### Decision · Program_Chairs · 2024-01-16

Reject